# Bacteria-on-a-bead: probing the hydrodynamic interplay of dynamic cell appendages during cell separation

Nora Sauter [1,2,3,5,7], Matteo Sangermani[4,6,7], Isabelle Hug[4], Urs Jenal [4] & Thomas Pfohl [1,2,3✉]

Surface attachment of bacteria is the first step of biofilm formation and is often mediated and coordinated by the extracellular appendages, flagellum and pili. The model organism *Caulobacter crescentus* undergoes an asymmetric division cycle, giving rise to a motile "swarmer cell" and a sessile "stalked cell", which is attached to the surface. In the highly polarized predivisional cell, pili and flagellum, which are assembled at the pole opposite the stalk, are both activated before and during the process of cell separation. We explored the interplay of flagellum and active pili by growing predivisional cells on colloidal beads, creating a bacteria-on-a-bead system. Using this set-up, we were able to simultaneously visualize the bacterial motility and analyze the dynamics of the flagellum and pili during cell separation. The observed activities of flagellum and pili at the new cell pole of the predivisional cell result in a cooperating interplay of the appendages during approaching and attaching to a surface. Even in presence of a functioning flagellum, pili are capable of surface attachment and keeping the cell in position. Moreover, while flagellar rotation decreases the average attachment time of a single pilus, it increases the overall attachment rate of pili in a synergetic manner.

[1] Institute of Physics, University of Freiburg, 79104 Freiburg, Germany. [2] Department of Chemistry, University of Basel, 4056 Basel, Switzerland. [3] Swiss Nanoscience Institute, 4056 Basel, Switzerland. [4] Biozentrum, University of Basel, 4056 Basel, Switzerland. [5]Present address: Bundesamt für Landwirtschaft, Eidgenössisches Departement für Wirtschaft, Bildung und Forschung, 3003 Bern, Switzerland. [6]Present address: Department of Circulation and Medical Imaging, Norwegian University of Science and Technology, 7030 Trondheim, Norway. [7]These authors contributed equally: Nora Sauter, Matteo Sangermani. ✉email: thomas.pfohl@physik.uni-freiburg.de

*C*aulobacter crescentus is a model system for the study of the bacterial cell cycle and cell polarity owing to its asymmetric life cycle[1]. The planktonic swarmer cell is motile and replication is temporarily arrested, it has a flagellum and several pili on one cell pole. A single flagellum is located at the apical end of one pole where it is used for swimming[2] and plays a role in surface attachment[3–6]. Swimming of bacteria in their surrounding environment is strongly dominated by viscous forces, whereas inertial forces can be neglected. Therefore, bacteria have developed different strategies for locomotion within viscous force-dominated environments, such as using rotating flagella, based on the consequence that only non-reciprocal motility patterns lead to a net displacement of the bacteria[2,7]. Many aquatic species, such as *C. crescentus*, follow a run-reverse-flick motility powered by a single flagellum, and yet are capable to swim in solutions with a wide range of viscosities[8,9]. The flagellum is composed of a large motor subunit, embedded in the cell envelop, and a long filament with a right-handed helical form that extends outside of the cell for 4–6 μm[10]. The swarmer cell changes its swimming direction by reversing the rotation direction of its flagellar motor[11]. In *C. crescentus* when the motor rotates clockwise (CW), it pushes the cell body forward[12]. To maintain torque balance, the cell body itself counter-rotates in a counterclockwise (CCW) direction. Every few seconds, the flagellar motor reverses its direction, forcing the flagellum to change its direction of rotation as well[13]. Consequently, the swimming direction is reversed with the flagellum now pulling the cell. Also, the counterrotation of the cell body changes from CCW to CW. The swimming velocity of the cells is around 40–60 μm/s and shows almost no difference for swimming forward and backwards[14]. *C. crescentus* possess polar tight adherence pili (Tad) belonging to the class of type IV pili, roughly arranged around the flagellum. Pili are dynamic appendages protruding outside the cell, which, among other functions, are used for surface attachment and motility[15]. Pili motors are embedded in the cell envelop and undergo cycles of retraction and extension by the disassembly and assembly of pilin subunits into a filament[16–18]. Pili filaments mainly act as anchors to attach and stabilize the cell upon a surface until a holdfast is secreted[19]. Motile swarmer cells of *C. crescentus*, upon surface contact or after a defined period of time, initiate the formation of an adhesive holdfast, which assures long-term, irreversibly attachment[1,6,20]. Attached cells start to grow a stalk and become replication competent, thus they are called stalked cells (or mother cells). Stalked cells can grow and keep generating new swarmer cells at the pole opposite to the stalk[21], which segregate from the mother cell harboring 2–3 Tad pili and a single flagellum on the free pole. After segregation, the daughter cell is released as a replication inert, motile swarmer cell, and the replication competent stalked mother proceeds with a new reproduction cycle. Owing to the fact that pili are used to anchor on a surface, while the flagellum is used for motility and exploration of new environments, it is easy to imagine that these two kinds of appendages act as natural counterparts whose coordination shapes the future of the swarmer cell.

Swarmer cells retain their motility for a defined period of time before undergoing a morphological transition, during which they shed the flagellum and their pili and expose an adhesive holdfast at the same pole[22]. However, when encountering a surface, swarmer cells are able to sense mechanical cues and immediately trigger holdfast formation resulting in firm surface anchoring[6]. This process can be visualized with attached predivisional cells, where the flagellated pole of the daughter side (future swarmer cell) is brought near the surface by applying flow[23]. The observation that rapid holdfast formation and surface attachment strictly correlates with flagellar obstruction, led to the idea that

the flagellar motor itself acts as mechanosensitive device[6]. Our previous work has revealed that both appendages, flagellum and pili, are central for surface sensing, resulting in permanent surface attachment[6,10]. Pili facilitate initial surface contact of the flagellated pole. This induces the diguanylate cyclase DgcB to produce the secondary messenger c-di-GMP. Higher concentrations of this molecule increase pili retraction activity, resulting in stronger surface contact, ratcheting up the c-di-GMP level until the holdfast machinery is activated, gluing the cell to the surface.

In this work, we investigated the role and the activity of the flagellum during cell separation. To investigate flagellar rotation during the division process with high temporal resolution, we established a bacteria-on-a-bead system by growing predivisional cells on colloidal beads. This experimental setup allowed us to precisely determine the onset of the flagellar rotation during *Caulobacter* division, detect and quantify changes in flagellar rotation and measure the forces generated and applied during this process. Moreover, we analyzed the interplay between the rotating flagellum and retracting pili shortly before predivisional cells separate in the vicinity of a surface. We show that bacteria-on-a-bead are not only an exciting tool to elucidate the hydrodynamics during cell separation, but also they provide an access to the dynamic processes of the bacterial response to surfaces.

## Results

**Swimming bacteria attached on beads**. In previous studies, we observed that the fate of swarmer progeny, namely to be able to rapidly form an adhesive holdfast and remain attached to the surface or to swim away, strictly correlates with the activity of the flagellum on the eve of cell separation[6]. To further analyze the behavior of the flagellum in the predivisional daughter cell, we designed a setup that allows the characterization of the dynamics of the flagellum as well as pili. We used the ability of *C. crescentus* to readily attach to nearly any available surface to generate predivisional cells attached to small polystyrene beads of different surface properties, thereby creating a "bacteria-on-a-bead" system suited to address our investigations (Fig. 1a, b). We discovered that cells attach straightforwardly onto colloidal beads, whereas the efficiency of attachment depends on the surface properties of the bead. We tested two differently functionalized bead surfaces, one with carboxylate surface groups and the other with primary amine surface groups, as well as non-functionalized polystyrene surfaces. The highest cell attachment rate was obtained for negatively-charged carboxylate-coated beads, followed by non-functionalized polystyrene beads and positively-charged amino-coated beads (Supplementary Fig. 1). First attachments of cells on beads were observed within minutes after addition of the beads to the cell suspension and the number of cells per bead increased over time. Interestingly, when the rotation of the flagellum started, the predivisional cells were able to swim when attached to a 1 μm bead. The experiments analyzing the swimming behavior of bacteria-on-a-bead where performed in microchamber devices with a height of 10 μm. In this limited space, the ~5–8 μm long bacteria-on-a-bead are essentially forced to swim horizontally. In addition, bacteria swimming close to a surface consistently swim parallel to it because they become "hydrodynamically trapped"[3,4]. The swimming behavior of bacteria in confined space is complex and it is determined by both cell body shape and flagellar architecture[24]. Crescent shaped bacteria, such as *V. cholera* and *C. crescentus* tend to swim parallel to surfaces when confined in close space[4,24]. Therefore, this setup creates a quasi-2D system that allows us to ignore the need of measuring negligible *z*-movements during microscopy observations.

A sequence of microscopy micrographs of swimming predivisional cells attached via a stalk and a holdfast to a colloidal bead is

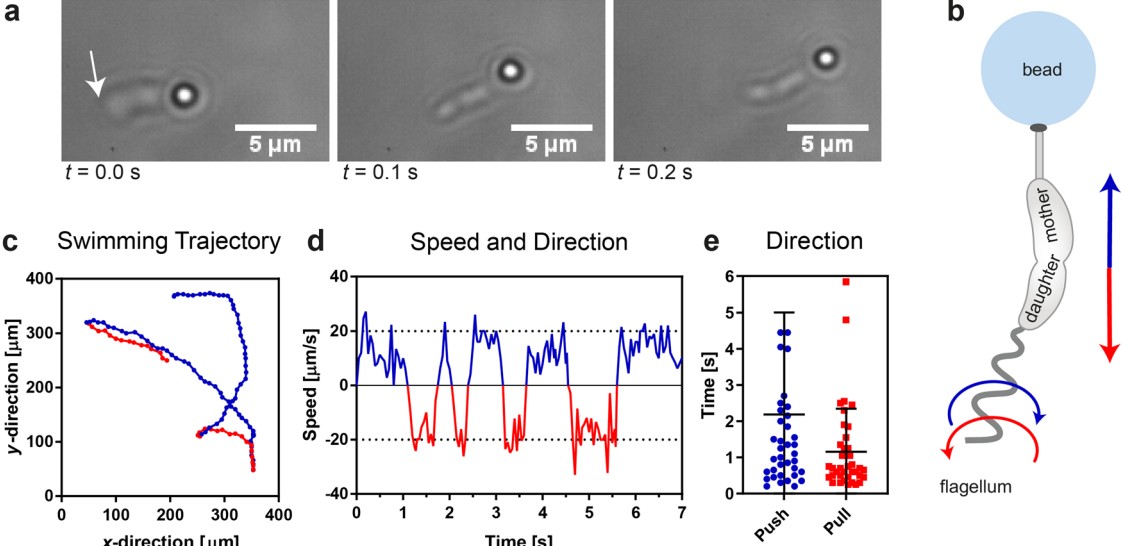

**Fig. 1 Swimming predivisional cells attached to a small bead. a** Consecutive bright field images of a swimming predivisional WT cell attached to a polystyrene bead with diameter 1 μm. The arrow indicates the flagellated pole. **b** Scheme of the predivisional cells attached to the bead (CW rotation of flagellum, causing forward swimming: blue. CCW rotation of flagellum, causing backward swimming: red) **c** Example trajectory of the swimming bacteria-on-a-bead. Forward swimming: blue, backward swimming: red. **d** Swimming speed of the cells attached to the bead in a thin microchamber device. Sign of speed values reflects swimming sequence: positive for forward swimming (blue) and negative for backward swimming (red). **e** Average duration of individual forward (push sequence) and backward (pull sequence) swimming sequences ($N = 3$, 39 measurements per direction, error bars represent ± standard deviation (SD)).

shown in Fig. 1a (Supplementary Movie 1). Because the cells can only be attached to the bead via the pole that forms the holdfast and stalk, the stalked side is closer to the bead while the flagellated side is at the free end (arrow indicating the flagellated pole in the first frame of Fig. 1a). This condition simplifies the determination of the swimming direction and therefore the rotation direction of the flagellum. When the flagellum rotates CW it generates a pushing movement and the bacteria-on-a-bead swims forward; vice versa when the flagellum rotates CCW it generates a pulling force and the bacteria-on-a-bead swims backward, hauling the bead. The plot line in Fig. 1c is an example of a trajectory tracking the position of a bacteria-on-a-bead over time, showing two forward (in blue) and two backward (in red) swimming sequences. The swimming speed and color-coded rotational directions are plotted in Fig. 1d. Backwards swimming (CCW rotation of the flagellum) of a bacteria-on-a-bead was slightly faster with a velocity of $(17.7 \pm 6.5)$ μm/s than forward swimming (CW rotation of the flagellum) with velocity at $(12.8 \pm 6.1)$ μm/s. We obtained a rough distribution of about 2/3 pushing and 1/3 pulling sequences; specifically, a pushing sequence lasted on average ~2.2 s and a pulling sequence ~1.2 s (Fig. 1e). This observed distribution is in good agreement with published data for free swimming swarmer cells[14], indicating that the switching behavior of the flagellum shows no major differences, whether they are predivisional or swarmer cells.

During the swimming period of the bacteria-on-a-bead, two different swimming-behaviors were observed. At the onset of the swimming the connection between the future stalked and swarmer progenies is stiff and the two halves of a predivisional cell do not change their positions relative to each other. This changes a few seconds before cell segregation, when suddenly the swarmer progeny starts to quickly rotate around its long axis while still being connected to the stalked mother, which is not rotating. The overall flagellar activity, from the onset of swimming until cell separation, lasted on average about $(99.8 \pm 57.3)$ s, with durations ranging from 40 s up to 3 min. Once the daughter cell is free to rotate, but is not yet

separated from the mother, we found that the rotation lasted on average for about $(10.3 \pm 7.6)$ s and ranged from 2 s up to 17 s (Supplementary Fig. 2).

**The trapped swimming bacteria attached to a bead.** In order to analyze the forces generated by the flagellum during cell separation, we conducted force measurements using optical trapping and imaging at high temporal and spatial resolution. In this approach, bacteria-on-a-bead were clamped by optical tweezers at the bead itself while lying inside a microfluidic device, where a strict no flow regime was established (Fig. 2a). The optical tweezers set-up allows for the immobilization and force measurements, while the bright-field microscope enables imaging (50–100 Hz) of the position of the bead as well as the predivisional cells. Using beads with a diameter of 3 μm, the attached cells are further away from the focused laser beam, therefore decreasing the amount of stray light that could inflict damages on the cells.

A series of bright field images of predivisional cells attached to a polystyrene bead is shown in Fig. 2b, c (Supplementary Movies 2 and 3). A single cell and a late-predivisional cell, with a recognizable segmentation between mother and daughter side, are attached to the bead. In panel b, the flagellum of the predivisional daughter cell is active, but the connection between the two sides is still firm. Predivisional cells in this state were able to change their position and move in a gyrating way, mostly in one direction (representative example shown in Fig. 2b). When the connection between the two sides of predivisional cells is softened, the gyration velocity is slowed down and the daughter side rotates around its long axis (Fig. 2c). The rotation of the daughter side is followed by the final cell segregation and the daughter cell swims away. After cell separation the bead with the remaining stalked cell shows only Brownian motion. These observations strongly indicate that the active flagellum of the predivisional daughter cell was the source for the gyrating motion. The finding that the propelling flagellum is the source of the gyration is supported by trajectories generated by cells lacking a flagellum ($\Delta flgDE$).

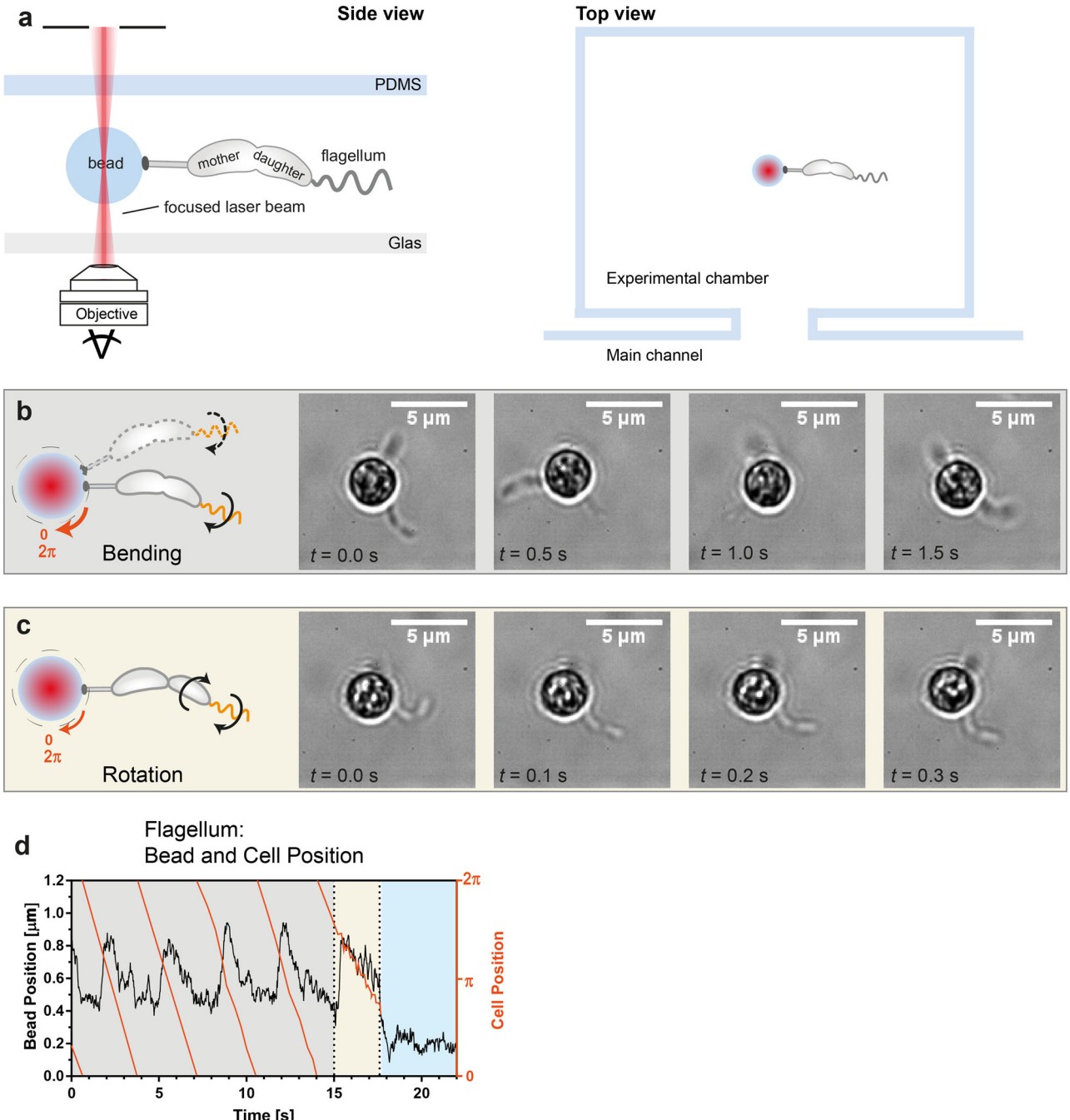

**Fig. 2 Predivisional cells attached to a bead in optical tweezers. a** Scheme of the experimental set-up (side and top view). The predivisional cells are attached to a polystyrene bead, placed in a microfluidic chamber and hold by optical tweezers. **b, c** Scheme and bright field images of two WT cells attached to a bead with a diameter of 3 μm in the optical tweezers. **b** The mother and the daughter sides of the predivisional cell are still firmly connected; the rotation of the flagellum induces a gyrational motion of the bead and the cell around the center of the optical tweezers. **c** The connection between the two sides of the predivisional cell is softened, the rotation of the flagellum causes a rotation of the daughter side around its long axis. The gyration of the predivisional cell proceeds albeit slower. **d** Position of the bead (black) and the cell (orange) during gyration of the cell (gray background), rotation of the cell around the long axis (yellow background), and when cell separation is completed (blue background).

Neither a significant displacement of the bead from the center of the optical trap nor a gyrating movement of the cell were observed in these control experiments and any recorded small motion was attributed to Brownian motion (representative example shown in Supplementary Fig. 3).

The propelling of the flagellum of the predivisional daughter cell did not only induce a gyration, but also a displacement of the bacteria-on-a-bead in the optical tweezers. The flagellum exerts gyrational and translational forces on the bead clamped inside the optical tweezers. The observed trajectories of the bead and the attached cells display the net of the effective gyrational and translational forces (Fig. 2d). From the gyration velocity of the bacteria-on-a-bead, the displacement of the bead from the center of the optical trap and the known trap stiffness $\kappa_{\text{trap}}$, we calculated the net force generated by the flagellum. We obtained a net rotational force of the flagellum of $F_{\text{rot}} \approx 0.3\text{pN}$ and a maximal

translational force of $F_{\text{trans}} = 1.2 - 1.5 \text{pN}$ for the trapped swimming bacteria-on-a-bead.

Analyzing several trapped bacteria-on-a-bead, the gyration stage as well as the rotation stage until cell segregation are of the same duration as for the free-swimming bacteria-on-a-bead: the gyration stage lasted on average about 200 s and the rotation stage on average less than 10 s (Supplementary Fig. 4). In the gyration stage, the trapped bacteria-on-a-bead showed in all experiments a preferred gyration direction with only rarely changing the direction of gyration, indicating that the gyration direction is not sensitive to the direction of rotation (pushing or pulling) of the flagellum. The finding that a change of the rotation direction of the flagellum does not in every case induce a change in the direction of the gyration may be attributed to the intricate geometry of the cells attached to the beads within the optical trap and the generated hydrodynamic forces. However, details of the switching of the rotation direction of the flagellum may be contained in the trajectories, which unfortunately cannot be deconvoluted in a straightforward way from the overall trajectories.

**Weakening of the connection between mother and daughter cells.** To further elucidate the rotation behavior of the daughter cells during cell separation and to reduce torque balance effects on the stalked cell, we grew predivisional cells with holdfast on glass substrates in microfluidic devices, which means that the predivisional cell is firmly attached to a non-rotatable substrate ("infinite counter mass", no torque balancing rotations). A few minutes before cell separation, we observed strong bending movements of the predivisional cells once again in response to starting the rotation of the flagellum (Fig. 3a and Supplementary Movie 4). During these bending movements, the two connected parts of the predivisional cell bent "as a whole" alternately towards the curvature and away from it. The maximal displacement of the flagellated pole could reach several micrometers, as one can see by tracking the x- and y-positions of the flagellated pole. In the example of Fig. 3a, the observed bending of the cell occurred mostly in the x-direction, as the cell was aligned nearly parallel to the y-axis. The bending movements lasted on average ~2 min for all analyzed predivisional cells, however, cells with movements up to 7 min could be observed (Fig. 3d). The bending movement was subsequently followed by the rotation of the predivisional daughter cells around their long axis, while still connected to the mother side (Fig. 3b). The rotation of the daughter side before segregation lasted <15 s in most cell divisions observed, although daughter cells could rotate up to 60 s (Fig. 3d). Moreover, it was also possible to determine the rotation direction of the cells owing to the distinct "crescent" shape of the cells. We obtained a distribution of about 3/5 pushing sequences – flagellum rotates CW – with a mean time of about 1.1 s and 2/5 pulling sequences – flagellum rotates CCW – with a mean time of about 0.8 s. The average rotation frequency for a CCW rotating cell was $(20.4 \pm 15.0)$ Hz and for CW rotating ones $(29.9 \pm 25.6)$ Hz (Supplementary Fig. 5). The individual rotation speeds varied from cell to cell, but for each cell observed the flagellum rotation was faster during CW mode than in CCW mode. After the rotation phase, the swarmer cells were released from the stalked cells and quickly swam out of the field of view (Fig. 3c).

The observed bending of the predivisional cells as well as the subsequent rotation of the daughter side around its long axis prior to the cell separation can be attributed to the incipient rotation of the flagellum. As long as the two halves of the predivisional cells are stiffly connected, the force generated by the flagellum causes predivisional cells to bend when they are attached to a substrate (Fig. 3a). Instead, when they are grown on a colloidal bead (Fig. 1) they can swim freely, or gyrate if the colloidal bead is trapped with optical tweezers (Fig. 2). As the cell cycle reaches a late stage, just before segregation, a transition from a stiff connection between the two cells to a soft connection can be observed. Now, the daughter cell side is free to rotate around its own axis, while still connected to the stalked cell side (Fig. 3b). This stage lasts several seconds and is terminated by the physical separation of the swarmer cell (Fig. 3c). To undergo the transition from a stiff connection uniting the two halves to a soft connection, the mechanical properties of the linking septum between the cells are changing. If the weakened connection were elastic, in response to the flagellum rotation, the counter-rotation of daughter side would cause an over-twisting of the elastic connection at the septum between the two cells (Fig. 3e). However, as one rotational sequence of the flagellum (either CW or CCW) reaches its end, we did not observe a decrease in the counter-rotation speed of the cells, which would be a strong indication of over-twisting. This could be explained by the formation of a liquid-like bridge between the two cells formed by the lipid layer of the outer membrane (Fig. 3e). In this case, the predivisional daughter cell would be able to rotate without over-twisting. The observed transition from a stiff cell connection to a soft one is the direct consequence of a change of the composition of the cell envelop of the gram-negative *C. crescentus* during cell division[25–27]. The cell envelope consists of an inner and an outer membrane, containing mostly lipoproteins and phospholipids, and a peptidoglycan layer in between giving the cell structural strength. After the separation of the rigid peptidoglycan layer, the two sides of a predivisional cell are solely connected by the outer lipid membrane and the daughter side is able to rotate around its axis owing to the conservation of angular momentum of the rotating flagellum.

**Competition and interplay of flagellum and pili at late stages of the cell separation.** *Caulobacter* swarmer cells are not only equipped with a flagellum, but harbor several pili at the same pole. To observe and analyze the interplay between flagellum and pili during cell separation, we performed experiments with a bacteria-on-a-bead held by optical tweezers. Once captured a bacteria-on-a-bead with optical tweezers, we moved the stage of the microchamber device relative to the optical trap, to place the bead close to the glass surface (at ~0.5 μm). This setup (Supplementary Fig. 6) and the "hydrodynamically trapping" of bacteria close to the surface[3,4] eliminate the non-trivial need to track the z-axis position. Although this used approach leads to an under-estimation of forces (~10%, Supplementary Fig. 6), the limited z-movements would provide marginal contributions to flagellar movements or pili pulling efforts, which do not alter the inter-pretation of the observed phenotypes.

In these experiments, we found that both flagellar rotation and pili retraction start 2–3 min before the completion of cell separation. An example, where a predivisional cell exhibited distinguishable phases of flagellum and pili activity, is shown in Fig. 4a. At first, there was a gyrational movement around the center of the trap generated by the rotation of the flagellum, just as in previous experiments without the involvement of a surface (i, gyration direction indicated by orange arrows). Suddenly, the gyration stopped, the bead showed a lateral displacement (ii, delocalization of the bead out of the center of the optical tweezers indicated by the yellow ring) and the cell shape was distorted. After a few seconds, the gyration of the bead was resumed (iii, indicated by an orange arrow). Because of the lateral displacement of the bead and the distortion of the cell shape, it can be excluded that a mere stop of the flagellar rotation caused the interruption in the gyration. Likely a sudden attachment of a pilus to the glass surface occurred. Upon attachment, the pilus arrests

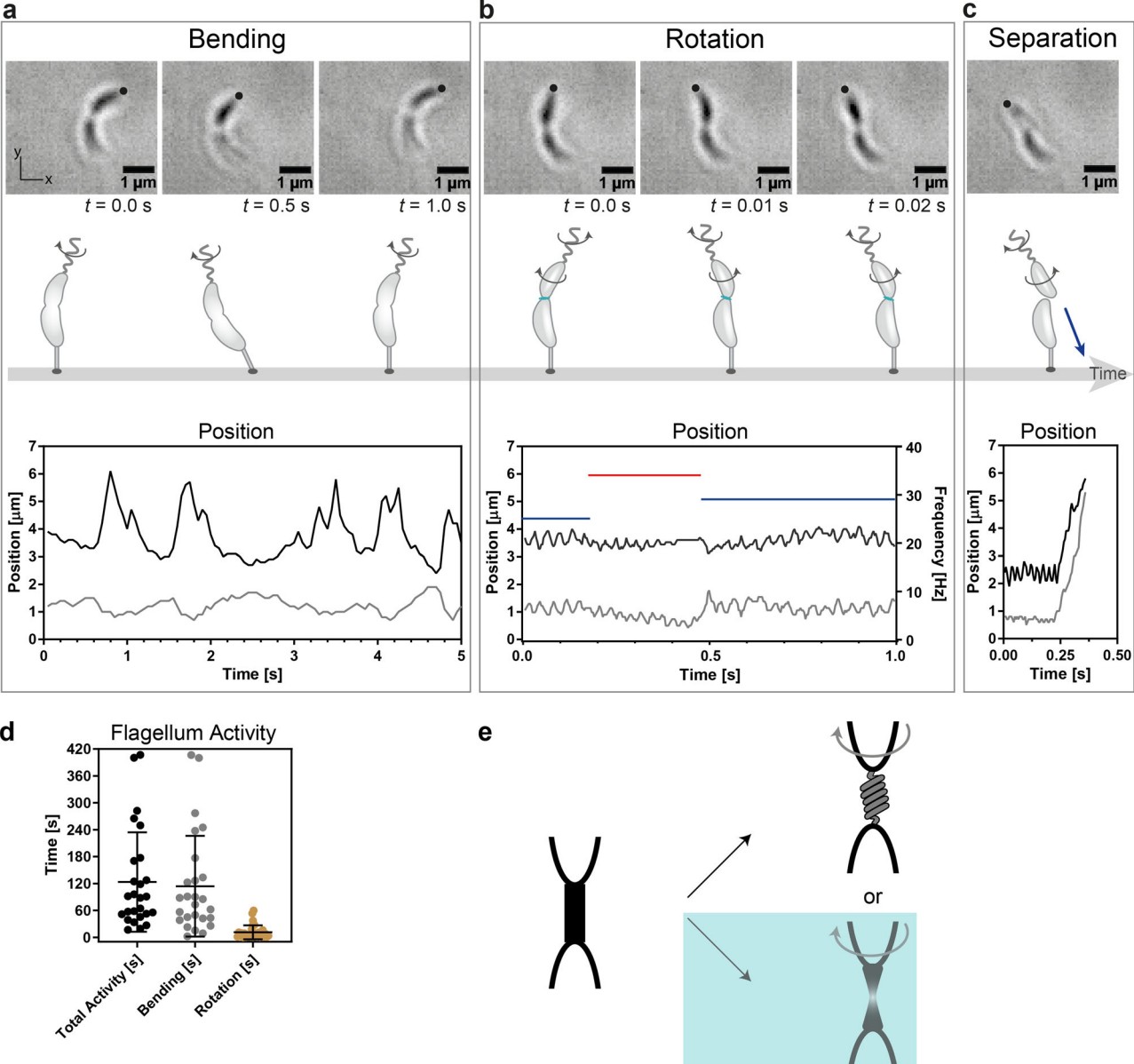

**Fig. 3 Bending and rotational movements of predivisional cells attached to glass surfaces ("infinite counter mass"). a** Micrographs and schematic representation of progression of cell separation and motion induced by the flagellum. Bending movement: start of flagellar rotation while the mother and the daughter side are still strongly connected. This rotation induces a bending of the predivisional cell. The strong displacements can be observed by tracking the x- and y-positions of the flagellated pole (black: x-direction, gray: y-direction). **b** Rotation movement: Flagellar rotation continues and the septum connecting mother and daughter side is softened. The rotation of the flagellum causes a counterrotation of the daughter cell while it is still connected to the mother cell. The oscillating displacements can be observed by tracking the x- and y-positions of the flagellated pole (black: x-direction, gray: y-direction). Rotation frequencies of freely rotating predivisional daughter cells of 25–35 Hz can be determined as well as switching of the rotation direction can be observed (blue: CCW rotation of cell body, red: CW rotation). **c** When cell separation completed, the swarmer cell is released from the stalked cell and is propelled by the rotating flagellum, which can be seen in the trajectories of the flagellated pole, now a swarmer cell – rotating motion followed by a ballistic motion (black: x-direction, gray: y-direction). **d** Duration of the overall activity from first observed bending movement until cell separation ($N = 26$), duration of the bending stage ($N = 26$), and of the rotation stage ($N = 29$). Error bars represent ± SD. **e** Schematic representation of two possible scenarios describing the softening of the septum connecting the two sides of a predivisional cell and the free rotation of the daughter side: over-twisting of the connection, which could not be observed, or the formation of a liquid-like bridge between the two sides (turquoise box), which is in agreement with our observations.

the gyration and maintains the cell in this position until the pilus detaches, whereby the cell immediately resumes its gyration. A representative trajectory of the bead, when both the pili and flagellum are active, is shown in Fig. 4b. When the flagellum is rotating freely it causes a gyrational movement (white area). When a pilus is attached to the surface it starts retracting and

pulling, causing a lateral displacement of the bead in the optical trap (blue area). In addition, we tracked the position of the cell over time (red line in Fig. 4b). When the flagellum is rotating freely and therefore dominating the displayed motion, the trajectory mainly plots as diagonal lines. When a pilus is attached to the surface, the cell remains in the same position for several

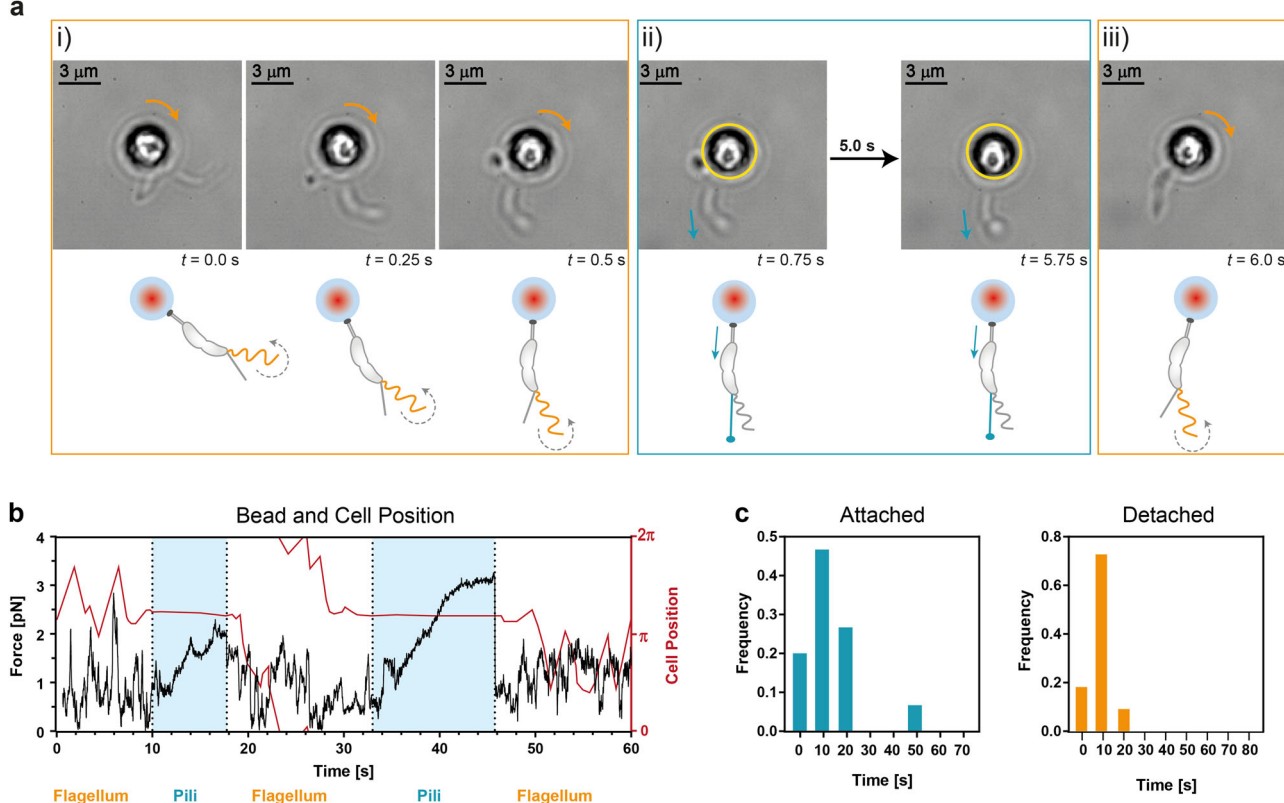

**Fig. 4 Interplay of flagellum and pili. a** Competition between flagellum and pili. (i) The active flagellum causes a rotational movement of the bead in optical tweezers (indicated by an orange arrow). (ii) Upon attachment of a pilus, the rotational movement is interrupted (indicated by a turquoise arrow). (iii) When the pilus is released, gyration is resumed (indicated by an orange arrow). **b** Flagellum and pili domination alternates. Typical trajectory of a predivisional cell with flagellum and pili attached on a colloidal bead in optical tweezers. When the flagellum is rotating freely, the bead shows a gyrational movement (white area). When the pilus is attached to the surface, the bead is dragged out of the optical trap in one direction (blue area). The active pili stabilize the cell for the duration of the attachment, resulting in a straight horizontal line. At the bottom of the graph, the dominating cell appendage is indicated. **c** Duration of individual pilus attachment cycles and period in between individual pilus attachment cycles ($N = 4$, 15 measurements for attachment cycles and 11 measurements for the period in between attachment cycles).

seconds, resulting in a horizontal line of the trajectory. During the assigned pilus attachment sequences, the bead shows a more directed movement compared to the flagellum associated sequences. Especially during the second pilus attachment sequence from 33 s until 46 s, the characteristic displacement of the bead is visible. This displacement resembles clearly the displacement trajectory of $\Delta flgDE$ cells[10]. The pilus slowly drags the bead out of the center of the trap, holds its position for some seconds and then the colloidal bead quickly moves back in the center of the trap. In between the two attachment cycles, the bead performed one full rotation. This demonstrates that during late predivisional stage, when both pili and flagellum are active, pili can attach and release several times without being impeded by the flagellar filament. Moreover, since a pilus is capable of interrupting gyrations in cells with fully functioning flagellum, the directional forces exerted by a pilus must be larger than the forces generated by the flagellum. A disassembly of pilin subunits is unlikely because we observed that gyration is immediately resumed without a transition regime. If it were otherwise, the restart of gyration would be slower and restricted. Thus, the release of the pilus is either accomplished by detachment of the pilus from the surface or breakage of the pilus filament. This hypothesis is also supported by the finding that the release speed is one order of magnitude larger than the retraction speed[10].

Analyzing the competing flagellum and pili activities before cell separation, we found that each pili-induced displacement of the bead lasted $(100.6 \pm 60.8)$ s and each flagellum induced displacement lasted $(230.9 \pm 207.4)$ s (Supplementary Fig. 7a). From the total observed time, displacement of the bead was caused in about 30% of the time by pili and the remaining about 70% by the flagellum. A pilus attachment cycle consists of four consecutive steps: attachment, retraction, hold, and release. The average duration of individual pilus attachment cycles is $(12.9 \pm 11.4)$ s, whereas the periods between two individual pilus attachment cycles are on average $(10.5 \pm 5.1)$ s (Fig. 4c). In comparison, the average duration of attachment cycles in cells without a flagellum ($\Delta flgDE$) is $(19.9 \pm 16.8)$ s[10] and the average period between two attachment cycles is $(23.4 \pm 22.0)$ s (Supplementary Fig. 7b). The presence of a flagellum decreases the duration of pili attachment events, indicating that the rotating flagellum generates an additional load and that it is continuously rotating even when a pilus is attached. Additionally, the flagellum could also dash against the attaching pilus, leading to a decreased attachment duration of the pilus. Surprisingly, the rotating flagellum increases the attachment rate of pili indicating that it supports pili contact with surfaces. This could be driven by the hydrodynamics of the flagellum, which brings the cell pole closer to the surface, therefore increasing the probability for a pilus to encounter the surface[28]. Elevated pili activity could also be driven by an increase of c-di-GMP upon surface sensing by the rotating flagellum[6,10].

## Discussion
In this work, we present a system that allows the direct observation of the dynamic and competitive interplay between the flagellum and pili in *C. crescentus* and the determination of the

acting forces shortly before and during cell separation. By attaching predivisional cells onto surface-functionalized polystyrene colloidal beads, imaging of the cells and simultaneous force measurements could be performed under physiological conditions. At the free pole of the predivisional cell, the flagellum as well as the pili are already active roughly 2–3 min before cell separation is completed. During the observed predivisional flagellum activity, two different successive stages can be distinguished. Firstly, the mother and daughter sides of a predivisional cell are strongly interconnected and the forces generated by the flagellum induce a motion of the predivisional cell as a whole, without any deformation at the septum and change in cell curvature. Owing to this, the predivisional bacteria-on-a-bead can swim with a velocity of up to 20 µm/s. Moreover, a switching of the swimming direction caused by the change of the rotational direction of the flagellum can be observed. As the cell approaches segregation, a short-term transformation occurs at the septum, whereby the connection between mother and daughter side is weakened such that the rotation of the flagellum forces the daughter side to rotate around its long axis while still connected to the mother side – no directional swimming of the bacteria-on-a-bead can be observed anymore. This phase lasts for a few seconds until cell separation is accomplished, the swarmer cell is released, and the stalked cell remains attached to the bead exhibiting only Brownian motion.

Consistent with previous reports[13], we observed roughly equal pushing and pulling rotational frequencies, which suggest that a similar torque is exerted during both CW and CCW rotation modes. However, we observed a small preference for CW rotations of the flagellum (pushing) before separation (Supplementary Fig. 5). Since both rotation directions could exert equally twisting of the septum, the preference for one direction is unlikely related to the need for separation of the two membranes.

One finds conflicting results regarding a bias between the pushing (CW) and pulling (CCW) swimming intervals in swarmer cells. Some works observed an increasing duration of the pushing swimming phase as swarmer cells climb ascending gradients, specifically an oxygen gradient[29], while others observed a constant ratio between pulling and pushing phases regardless of gradients[30]. However, we observed that our bacteria-on-a-bead spend twice as much time pushing than pulling (Fig. 1c). Moreover, we found that predivisional cells have longer sequences for both pushing and pulling than reported in swarmer cells of *C. crescentus*[30]. Although our cells were not placed in a chemical gradient, these results suggest that motor rotations in predivisional cells may works slightly differently. Prolonged rotational phases in predivisional cells could favor bending and twisting of the septum and in turn facilitate cell membrane separation. The increased duration of the sequences could be achieved by locking the chemotaxis system of predivisional cells. However, as yet, there are no studies to elucidate if and how chemotaxis works in predivisional cells.

The predivisional activities of the flagellum and pili lead to a fascinating interplay of the two kinds of cell appendages that are pivotal for surface colonization behavior in bacteria. Naturally, one could consider that these two cell appendages act antagonistically when cells are in the vicinity of a surface. While pili anchor cells on the surface and keep them in a fixed position, the rotary flagellum should act against this and promote cell dispersal. However, our study suggests that the interactions between pili and flagellum are more complex and possibly synergistic. We found that the pili of predivisional *Caulobacter* cells can attach to the surface even when the flagellum is actively rotating. Because the attachment force of a pilus exceeds the forces generated by the flagellum, it is reasonable to assume that cells attached to the surface via their polar pili use the rotating flagellum to sense

mechanical cues as was previously proposed[6,10]. The observations that a rotating flagellum decreases the attachment duration of pili but increases the rate of pili attachment is sensible, as swarmer cells have a finite time to probe surfaces and to make informed decisions about adhering or retaining their motile state.

A rotating flagellum may also facilitate active cell separation during division. As soon as the division septum is formed, an active flagellum enables cell bending and twisting. At the late predivisional stage, bending and twisting due to the flagellum could tally to restrict the cell septum together with the coiling forces generated by the cytoplasmatic complex, thus the flagellum activity could accelerate and speed up terminating the separation. To explore this aspect further, an approach could be taken that specifically alters or prevents the phenomenon of fluid-like bridging, such as modifying the fluidity of the bacterial membranes. For example, this modification can be achieved by thinning and weakening of the cell walls by using antibiotics (i.e. β-lactams) or alternatively by using strains with mutations in the peptidoglycan synthesis.

Our findings suggest that in *C. crescentus* the flagellum has low or absent surface attachment capability, yet the onset of the flagellar rotation and the pili activity before the completion of cell separation assists cellular surface attachment. As the flagellum is known to be involved in surface sensing[6], the active flagellum helps the predivisional daughter cell to sense the surface and initiate holdfast formation before cell separation is completed, hence increasing the probability of surface attachment. Moreover, the active flagellum is also capable of bending the attached predivisional cell, therefore the surface sampling area of the flagellum is increased. Together with the active pili, the combined efforts could allow a predivisional cell with its flagellated pole, which is initially too far away from a surface for pili to attach, to bring its pole closer to the surface, therefore improving pili attachment efficiency during initial as well as later stages of biofilm formation (Fig. 5).

Our bacteria-on-a-bead approach is a valuable tool for reducing phototoxicity in optical tweezers experiments, or more excitingly to test the impact of adhesion and hydrodynamics of cell appendages on cell separation, surface attachment and colonization. Although not trivial, it is conceivable to create a "bacteria-on-two-beads" system. Exploiting holdfast secretion and its powerful adhesive properties, a strain that is prone to early holdfast secretion could be induced to prematurely bind to a second bead approached to the flagellated pole. In such an arrangement, the bacteria would sequentially bind the two ends of the predivisional cell with two separate beads. Multi-beam optical tweezers could then be used to test mechanical parameters of the cells at late stages of the division, by synchronized pulling and/or pushing movements of the two beads. Moreover, the existence of a liquid-like bridge during softening of the septum could be directly measured by performing coupled twisting and pulling motions of the two beads.

As a tool and toy model, our system may also represent a possible "infinite self-powered" and load-selecting biohybrid micro-transporter[31,32]. Bacteria-on-a-bead can represent a very interesting micro-transport and delivery system. Being self-powered by continuous binary fission and capable to follow chemical gradients with high accuracy, the bacteria-on-a-bead can deliver precisely bound loads to locations within natural and synthetic microfluidic systems.

## Methods

**Cell preparation.** Cells were grown overnight in peptone yeast extract[33] (PYE, in-house media kitchen, see Supplementary Note 1 for composition) under agitation at 30 °C. To attach cells to colloidal beads, the overnight culture was diluted 1:10 with fresh PYE and put back at 30 °C for 2 h (exponential phase cultures). The cells

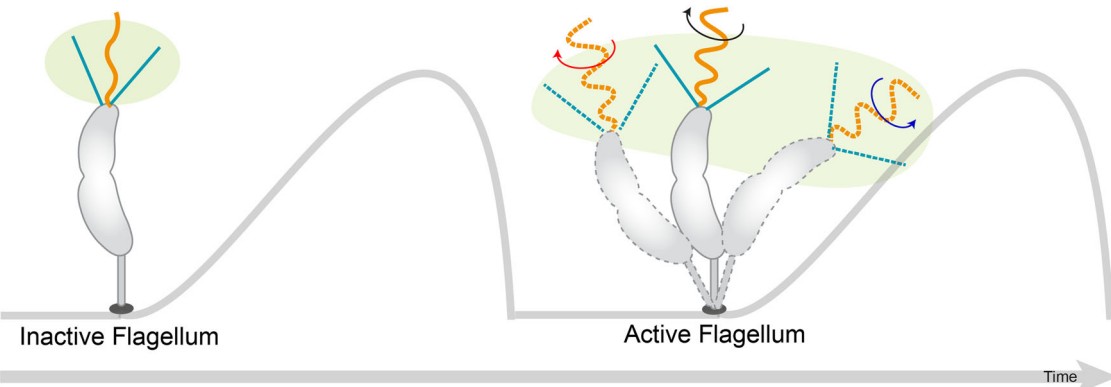

**Fig. 5 Surface sampling area of the predivisional daughter cell before (left) and during (right) rotation of the flagellum.** Because the rotating flagellum can bend the predivisional cell, the surface sampling area is increased when an active flagellum is present (surface sampling is indicated in light green).

were then further diluted 1:2 with PYE and the appropriate amount of polystyrene bead solution (Polysciences, Inc., Warrington USA) were added and well mixed by pipetting. For the beads with a diameter of 3 μm, 25 μl of bead solution was added for each ml of cell suspension (final concentration: $1.7 \times 10^9$ beads/ml); for the beads with a diameter of 1 μm, 1 μl of bead solution was added for each ml of cell suspension (final concentration: $4.6 \times 10^{10}$ beads/ml). The cell-bead mix was incubated for 2 min, then a small amount was placed on a coverslip and the numbers of attached cells per bead were checked. When the desired number of beads of about 1–2 attached cells was reached, the cell suspension was diluted 1:1 with fresh PYE and injected into a microfluidic device. The tubing was removed and the device was submerged in PYE to create a strict no-flow environment and to avoid evaporation[34]. Recording of the cells was started ~2 h after injection and stopped ~8 h after injection. At later time points, the device was overgrown with bacteria. To attach cells onto the surface of a microfluidic device an overnight culture of cells (roughly $1.4 \times 10^9$ cells/ml) was diluted 1:1 with fresh PYE and injected into a microfluidic device. The flow was stopped, and the cells were left to attach onto the surface of the microfluidic device for 2–3 minutes. Planktonic, non-attached cells were flushed by injection of fresh PYE at a flow rate of ~0.1 μl/s. This also supplied the attached cells with fresh media. After 20–30 min, the flow was stopped, the tubing removed and the PDMS (polydimethylsiloxane) device as submerged in PYE.

**Strains used**. The strain NA1000 $hsfA^+$ is referred to as wildtype (WT)[6]. It has a flagellum and Tad pili. The flagellum changes its rotation direction regularly. The strain NA1000 $hsfA^+$ $\Delta flgDE$ ($\Delta flgDE$) does not have the outer parts of the flagellum due to the deletion of flagellar hook genes[6]. All used strains form holdfasts, the ability to irreversibly attach to surfaces was indispensable for the experiments.

**Microfluidic devices**. Standard photolithography and PDMS casting were performed as previously described[35]. After covalently binding the PDMS replica to a glass slide, a small plastic ring was dipped into liquid PDMS and placed to encircle the mold. Devices were thermally aged to render the surface inside the device more hydrophilic and to get rid of any remaining low molecular weight polymer chains[36,37]. The devices were placed on a hotplate for 4 h at 150 °C and stored at 80 °C until use.

To prevent sticking of the polystyrene beads to the surface, the microfluidic devices were coated with 1 mg/ml bovine serum albumin (BSA, Sigma) dissolved in deionized water. After rinsing the device with PYE, the BSA solution was pumped through the device for several minutes (pump module: neMESYS low pressure syringe pump V2, 14:1 gear; Cetoni GMBH), followed by a rinsing step with PYE to prepare the cell injection.

**Optical tweezers and imaging**. All experiments were performed on a custom-built bright field microscope combined with optical tweezers. The optical tweezers set-up consists of a tunable laser diode (1 W, 830 nm, LD830-MA1W - 830 nm, Thorlabs) and a lens system to collimate, align, and expand the laser beam to overfill the back aperture of the microscope objective (×60 water, 1.20, UPlansApo, Olympus). The laser beam was focused through the back aperture of the objective. The experiments were performed in position clamp mode at a constant laser power. The optical tweezers were calibrated via thermal fluctuation calibration.

A high-speed camera (Phantom Miro eX4, Vision Research) was used for all recordings. Cells attached to the cover-slide glass surface were recorded at 200–300 Hz, cells attached to 3 μm beads at 50–100 Hz and cells attached to 1 μm beads at 10–20 Hz. The maximum frame rate inversely correlates to the number of pixels imaged; therefore, a lower frequency of imaging is imposed when choosing a larger field of view. The larger field of view was needed for larger scaled objects (cells attached to 3 μm beads) and for objects that could freely move (cells attached to 1 μm beads).

**Data analysis**. The recorded images of cells attached to a solid surface were analyzed manually to determine the duration of the flagellar activity and the rotation direction in Fiji[38]. The recorded images with cells attached to 1 μm beads were analyzed using the Fiji plugin "Manual Tracking"[39] to track the position of the beads. The moments of onset of the flagellar rotation, the softening of the connection between the two separating cells and the moment of cell separation were determined using this plugin.

The position of the 3 μm bead hold by the laser was determined via an in-house developed MATLAB (MathWorks) app "Bead Tracker"[40]. The script determined the position of the bead for each frame. From analyzing the trajectory, we determined the onset of flagellar rotation and pili attachment as well as retraction and release events. From these measurements, the attachment duration and efficiency were calculated. Using the data from analysis with the "Bead Tracker" software, we generated graphs that were used to determine and display the position of the cell for each frame. With these scripts, it was possible to determine the onset of the flagellar rotation, angular velocity, reversion of the rotation direction, the softening of the connection between the two cells, and the moment of cell separation.

**Statistics and reproducibility**. All quantitative data are presented as average values with error bars which represent ± standard deviation (SD). The average values and SD were assessed using Matlab and GraphPad Prism. Sample sizes are stated in the manuscript for each graph when applicable. Measurements were generally obtained from at least three different samples or experiments.

**Reporting summary**. Further information on research design is available in the Nature Research Reporting Summary linked to this article.

## Data availability
The source data for graphs in the main figures and supplementary figures are provided in the Supplementary Data 1 and 2, respectively.

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

## Acknowledgements

We would like to thank Benjamin Banusch, Siddharth Deshpande and Axel Hochstetter for their experimental support and fruitful discussions during early stages of the project. This work was funded by the Swiss Nanoscience Institute in Basel, Switzerland (SNI PhD graduate school, Project P1302). M.S. acknowledges funding from SNF grant P400PM_194492 and U.J. from SNF grant 310030B_147090 by the Swiss National Science Foundation.

## Author contributions

N.S., M.S., I.H., U.J., and T.P. designed the research. N.S. and M.S. performed the experiments. N.S., M.S., and T.P. analyzed the data. N.S., M.S., and T.P. wrote the paper with support of all authors.

## Funding

## Competing interests

The authors declare no competing interests.
