## [Peer Review File · Communications Biology]

Reviewers' comments:

Reviewer #1 (Remarks to the Author):

This is a beautiful paper. I had very little to say about it, initially, but then I thought about my own experience, if I would have done it myself (which obviously it is not the case - this is much better than what I would have been able to do). So, few, possibly important suggestions came to my mind:

1. First, in the essentially 2D images, i.e., in the movies, it would be instructive to provide the actual trajectories of the "bead-on-a-cell" system - although this could be deceiving.
 2. Furthermore, it would be possible to "map" the trajectories of the "bead-on-a-cell" system in 3D by confocal imaging. I know this exercise is not trivial, but it is worth it, and it will greatly increase the quality of the contribution.
 3. Of course, this would be much more difficult (isn't it?) for experiments using the optical trap, but difficulties aside, it would represent the reality - the bacterial system is not moving "around" the bead in a plane, but in 3D. However, once the movement is/would be mapped in 3D, one could "planarise" the motion and present it as it would have been in a plane.
 4. With this approach, or perhaps even without it, it would be very useful to estimate the forces involved. I guess that this, while not trivial, would be possible, precisely as the authors are using optical trapping.
 5. Finally, it would be useful to comment, even speculatively, and more, on what is the impact of this study. While it is argued, I think correctly, that the methodology would be useful for a number of other studies, I do not feel that the point regarding the usefulness of the knowledge gain had been made (forcefully enough).
- Again, congratulations on a very nice contribution.

Reviewer #2 (Remarks to the Author):

I read this manuscript by Sauter et al with great pleasure. It is a great piece of work dissecting from multiple directions the effect of flagellum and pili mediated forces in the asymmetric cell division of *C. crescentus*. The paper is extremely clearly written, explains a very complex process of cell division in a step by step manner and supplements with clear and informative figures. On the technical side, I want to highlight several complementing experimental settings: bacteria-on-bead, bacteria-on-bead in optical trap, bacteria attached to a surface, and, finally, combination of optical trap and surface interaction. From a biological point of view, this wealth of data (mostly force data, and tracking data) converges on the very nice observation of the liquid-like lipid bridge between the mother and daughter cell during the final stages of cell division. It further clarifies the coordinated operation of pili and flagella during the division process. While I do think that this manuscript is nice and round finished piece of work, to finally sort out all biological details it must have a continuation in the future, which I hope to see soon published from this group. Thus I fully support the publication of this manuscript, after the authors respond to a series of minor comments below.

In chronological order:

Line 71 and below in the text: cell appendices should probably be appendages

Line 76 which assure -> assures

Figure 1D. Velocity with respect to which axis is shown? Actually speed is the absolute value of velocity, and velocity has the direction, so both the caption and the y axis are problematic. I guess what is shown is just a an x or y coordinate of cell velocity?

Relatedly - can authors provide the dimensions of the channel - its z-dimension? Is the swimming of beads is 3D, or quasi 2D? I looked but couldn't find immediately.

Line 233, while it is clear how to distinguish the phases of forward and backward swimming, how was the phase of daughter cell rotation identified and quantified?

Lines 334-343. The idea of liquid-like bridge is fascinating, and indeed physical data (forces, tracking) do point to this mechanism. While I'm not insisting on doing this in the revision of this ms, can the

authors at least suggest how this mechanism could be tested? Can one perturb membrane composition to abrogate this effect? Can one directly rotate two cells with respect to each other in different phases of division and show the phase transition of the torque from a certain value to nearly zero? A discussion of possibilities would be great.

Line 367. In the final setup the authors hold the bead so that the daughter cell has a chance to grab on the surface with pili, or, in other words, flagellum and pili face the surface. However, in the natural setting it seems that the daughter cell points its flagellum and pili away from the surface. Does it mean that pili are not too important for division? Did authors see pili binding in the setting when cells were attached to the surface?

Relatedly, when cells face with pili towards surface (when held by optical tweezer) but lack flagella, will they still be able to separate?

Reviewers' comments:

General reply:

Thank you very much for the suggestions and comments. In the past, we had considered the suggestions that have been proposed by Reviewer 1. However, as highlighted by the reviewer her/himself, these are not trivial. Performing a full 3D tracking in our case (the major suggestion by Reviewer 1) would have been a display of technical skills, but one that is very complicated, time consuming and would not have contributed to the interpretation of our findings. Most importantly, we overcame these issues by creating a specific setup in our described work, such to circumvent the need to measure the z-axis displacement. However, considering the comments, we see that this was not clearly enough described in our manuscript. Thus, we have amended with the addition of one Supplementary Figure and improvements in the text to be more evident.

Both Reviewers have also highlighted that we do not push enough the potential applications and uses of our setup, namely the bacterial-on-a-bead. We were overly cautious in our text and we find their comments encouraging. Thus, we amended the Conclusions in this aspect.

Reviewer #1 (Remarks to the Author):

This is a beautiful paper. I had very little to say about it, initially, but then I thought about my own experience, if I would have done it myself (which obviously it is not the case - this is much better than what I would have been able to do).

We are happy about the very positive assessment of the reviewer about our paper and thank her/him for her/his suggestions.

So, few, possibly important suggestions came to my mind:

1. First, in the essentially 2D images, i.e., in the movies, it would be instructive to provide the actual trajectories of the "bead-on-a-cell" system - although this could be deceiving.

We updated and overlay a track with a representative movie of a bacteria-on-a-bead system swimming in our device, similar to the representative trajectory of Figure 1C. We understand the warning of Reviewer 1 that an addition of trajectory could be potentially confusing for some readers. However, the setup can be considered as quasi-2D system (see below), and the lack of 3D tracking to be inconsequential.

Free swimming of bacteria-on-a-bead was performed in devices with a height of 10 μm . This information has been added in the text, which was missing so far. With a bead of $\sim 1 \mu\text{m}$, the bacteria-on-a-bead system has total length of about 5-8 μm , plus a flagellum of up to 10 μm (not visible in light microscopy). In this limited space, swimming bacteria-on-a-bead essentially cannot orient vertically to any significant degree, and in fact we never observed swimmers doing so. Indeed, as previous publications have shown [3,4], bacteria swimming close to a surface consistently swim parallel to it because they become "hydrodynamically trapped". By swimming between two surfaces with a 10 μm gap, the system is effectively inducing to swim parallel to the xy-plane (our plane of observation), further limiting z-displacements during swimming.

Without the z-axis we may slightly underestimate the swimming speeds, but we believe this to be very marginal. Considering that we use free swimming to identify pulling/pushing sequences, the additional information from the z-axis would not have contributed to improve the interpretations of our observations.

For clarification and emphasis, we have add lines 182-188.

2. Furthermore, it would be possible to "map" the trajectories of the "bead-on-a-cell" system in 3D by confocal imaging. I know this exercise is not trivial, but it is worth it, and it will greatly increase the quality of the contribution.

Three dimensional tracking of moving objects using confocal fluorescence microscopy is a very sophisticated approach, e.g. use and access to a spinning disc confocal fluorescence microscope. Moreover, we used fluorescence markers in our settings (not included in the submitted work), but combining fluorescence with our microscopy and optical tweezers setup created excessive phototoxicity for *C. crescentus* cells, resulting in arrested growth. Even more resilient bacteria like *E. coli* could not survive high frequency frame rates confocal imaging in combination with optical tweezers.

3. Of course, this would be much more difficult (isn't it?) for experiments using the optical trap, but difficulties aside, it would represent the reality - the bacterial system is not moving "around" the bead in a plane, but in 3D. However, once the movement is/would be mapped in 3D, one could "planarize" the motion and present it as it would have been in a plane.
4. With this approach, or perhaps even without it, it would be very useful to estimate the forces involved. I guess that this, while not trivial, would be possible, precisely as the authors are using optical trapping.

We reply to suggestions 3 and 4 at once and would like to draw your attention also to reply to suggestion 1, which partly addresses these suggestions as well.

It is in principle possible to determine the generated forces from a non-trivial analysis of "planarized" motions of the bacteria-on-a-bead. Although we are able to "planarize" the motions of bacteria-on-a-bead due to our setup (see above) and to measure the velocities/speeds, we are not able to image the dynamics of the moving bacteria constructs with high precision and resolution using our optical setup – one is not able to image the flagellum and pili by bright field microscopy. The exact details of the dynamics and shape changes of bacteria-on-a-bead are crucial in order to determine the friction factor and thus the (exact) forces. Therefore, we have chosen the optical tweezers settings for our experiments in order to more directly measure the generated forces of the bacteria-on-a-bead. Fortunately, we have observed that a trapped bacteria-on-a-bead with an active flagellum shows a gyrational motion, which can be used to characterize the competition and interplay of flagellum and pili in the later described experiments. We have no evidence in our experiments that the "moving" around the bead (gyrational motion) has a direct impact on the properties of the flagellum.

In case of the experiments on the competition of flagellum and pili, once again and thanks to our setup, we can obviate the need to track the z-axis position and the marginal contributions from flagellar movements or pili pulling efforts in the z-axis do not alter the interpretation of the phenotypes observed. As sketched in SI Figure 6, by measuring the forces of pili we only underestimate the effective force of pili by about 10 %.

We admitted in the text that we are underestimating a little our measurements, but we believe we did not elaborate sufficiently. To address the concerns on 3D tracking and force measurements, we have added SI Figure 6 that depict the bead-surface-cell-pili system and estimate of dimensions involved. We also have amended the text to be more explicit and explain how our setup obviate the need of tracking z-axis displacements (lines 356-362).

5. Finally, it would be useful to comment, even speculatively, and more, on what is the impact of this study. While it is argued, I think correctly, that the methodology would be useful for a number of

other studies, I do not feel that the point regarding the usefulness of the knowledge gain had been made (forcefully enough).

We have commented more on potential applications and uses of our setup, such as “two beads setup” and development of micro-transporters (addition of line 504-517)

Again, congratulations on a very nice contribution.

Thank you very much!

Reviewer #2 (Remarks to the Author):

I read this manuscript by Sauter et al with great pleasure. It is a great piece of work dissecting from multiple directions the effect of flagellum and pili mediated forces in the asymmetric cell division of *C. crescentus*. The paper is extremely clearly written, explains a very complex process of cell division in a step by step manner and supplements with clear and informative figures. On the technical side, I want to highlight several complementing experimental settings: bacteria-on-bead, bacteria-on-bead in optical trap, bacteria attached to a surface, and, finally, combination of optical trap and surface interaction. From a biological point of view, this wealth of data (mostly force data, and tracking data) converges on the very nice observation of the liquid-like lipid bridge between the mother and daughter cell during the final stages of cell division. It further clarifies the coordinated operation of pili and flagella during the division process. While I do think that this manuscript is nice and round finished piece of work, to finally sort out all biological details it must have a continuation in the future, which I hope to see soon published from this group. Thus I fully support the publication of this manuscript, after the authors respond to a series of minor comments below.

In chronological order:

Line 71 and below in the text: cell appendices should probably be appendages

Line 76 which assure -> assures

We changed it to the correct terminology in the text and corrected the typo.

Figure 1D. Velocity with respect to which axis is shown? Actually speed is the absolute value of velocity, and velocity has the direction, so both the caption and the y axis are problematic. I guess what is shown is just a an x or y coordinate of cell velocity?

In Figure 1D we do not show speed of one axis, but absolute speed in the *xy*-plane. However, we understand the source of the confusion, because the plot shows speed as a line plot with negative values. Indeed, we could have kept all speeds as positive values and simply colored the different swimming phases. However, we thought to apply the sign to further highlight the different sequences.

We have amended the caption of Figure 1 and changed the wording and title from *velocity* to *speed*.

Relatedly - can authors provide the dimensions of the channel - its z-dimension? Is the swimming of beads is 3D, or quasi 2D? I looked but couldn't find immediately.

Free swimming of bacteria-on-a-bead was performed in devices with a height of 10 μm . We amended the text and added such information, which were previously missing (lines 182-188). We consider our setup as a quasi-2D system because as shown by previous publication [3,4] bacteria swimming close to a surface become "hydrodynamically trapped" and swim parallel to the *xy*-plane. The setup effectively induces the bacteria-on-a-bead system to swim parallel to the *xy*-plane, limiting z-displacements (see also answers to Reviewer 1).

Line 233, while it is clear how to distinguish the phases of forward and backward swimming, how was the phase of daughter cell rotation identified and quantified?

Observing at low frame rate it is possible to differentiate early and very late predivisional phase. At late stage, the swarmer cells spin very quickly around its own axis, while at the same time the mother cell appear static. Also CW and CCW spinning display slightly different amplitude. A switch in rotational direction appears as a very short stop and the cell position, mother included, drift more than usual.

The latter is a symptom of the tumble-flick, typical of bacteria and that is pivotal to reorient cells randomly (clarification in line 220-221).

Lines 334-343. The idea of liquid-like bridge is fascinating, and indeed physical data (forces, tracking) do point to this mechanism. While I'm not insisting on doing this in the revision of this ms, can the authors at least suggest how this mechanism could be tested? Can one perturb membrane composition to abrogate this effect?

Can one directly rotate two cells with respect to each other in different phases of division and show the phase transition of the torque from a certain value to nearly zero? A discussion of possibilities would be great.

We were cautious in adding speculations, but the comments suggest that this may be important for a better understanding of potential readers. Therefore, we have amended the conclusions to include a more detailed discussion of possibilities (addition of lines 480-484 and 504-512).

Line 367. In the final setup the authors hold the bead so that the daughter cell has a chance to grab on the surface with pili, or, in other words, flagellum and pili face the surface. However, in the natural setting it seems that the daughter cell points its flagellum and pili away from the surface.

Does it mean that pili are not too important for division? Did authors see pili binding in the setting when cells were attached to the surface?

Previous works, including ours, have experimented with *C. crescentus* cells under flow conditions. The bacteria have the natural ability to spread from a single cell and form local colonies under flow conditions, thanks to pili and their curved cell shape [23]. This means about 50% of wildtype cells attach next to their mother cells under flow, while the remaining cells are dispersed to establish new colonies [6,10]. E.g., in the natural habitat of pots and rivers with always (small) flow of water which would bend the stalk and help cells to reach the nearby surface. Also, from a biomechanical point of view, flagellum rotations can help by potentially bringing the cell close to surface and spreading pili [4]. This gives pili the chance to feel and attach on the nearby surface [10]. In our setting of stalked cells on flat glass surfaces under no flow conditions, almost no pili binding events could be observed.

Relatedly, when cells face with pili towards surface (when held by optical tweezer) but lack flagella, will they still be able to separate?

They attach, pull, and separate like any other cell, albeit never displaying any swimming behavior as it is shown in reference [10].

REVIEWERS' COMMENTS:

Reviewer #1 (Remarks to the Author):

1. Referring strictly to the added text (yellow), in no particular order

a. "Data Availability Statement. The datasets generated during and/or analyzed during the current study are available from the corresponding author on reasonable request." I am not at all clear what is "reasonable", or for that matter unreasonable, request. I strongly suggest that all the data from which the findings are derived is directly available, without any need for (un)reasonable request. I believe that this is not unreasonable, but common.

b. "Swimming bacteria on beads". Rephrase. It would mean that bacteria swim ON beads. One way would be to declare a composed noun, e.g., "bacterium/a-on-bead/s", or bacterium@bead, etc.

c. "surface consistently swim parallel to it because they become "hydrodynamically trapped" [3,4]." I believe that this is "scientific hand waving". There is serious literature, in particular from Henry Shum (who spent all his PhD @ Oxford looking into this). He found that for mono-flagellate bacteria, which is the case here, the ratios of the cell width/cell length, and cell length/flagellum length, dictate if bacteria moved either as wall accumulator, or wall escaper, or parallel (but away) from the wall. This work was later extended (PNAS) to "all" bacterial architectures, to find out that largely this classification holds true even for more complicated architectures. It would be useful to have a discussion about it, or at the very least let the reader go deeper into Shum's (and others) work.

d. Minor grammar double-check: "Once captured a bacteria on-a-bead with optical tweezers, we...". I guess it's "Once WE captured...", or "Once a bacterium (singular, if "a") is captured...". Also, later, "Although this used approach..." does not sound like English. I think that a professional scientific editing is needed.

2. Referring to the Rebuttal letter. It happens that I am not convinced that the z-staking would be that difficult. Be that as it may be, if something is difficult, it does not mean that it is not necessary (often the opposite is true), but apparently this is the authors' line of defence. If experiments are difficult, not achievable, some effort must be made to qualify the observed facts, and possibly extract inference from whatever data were available. I find that a more thorough, theoretical at least, discussion is needed here.

3. Recommendation. I think that at least the easy suggestions can be done without problems. The more difficult one, i.e., discussion about trajectories in 3D, could be "just" better substantiated. With these done, I guess that paper can be published.

Reviewer #2 (Remarks to the Author):

The authors replied to all my comments and im gladly recommending this very nice manuscript for publication!

Point-by-point response to Reviewers' and Editors comments:

Thank you very much for the comments and your congratulations on an excellent paper.

We ask you please include discussions about 1) the swimming behavior close to a surface and 2) trajectories in 3D.

We have included a discussion on these points – see for more details our answers to the reviewers.

Reviewer #1 (Remarks to the Author):

1. Referring strictly to the added text (yellow), in no particular order

a. “Data Availability Statement. The datasets generated during and/or analyzed during the current study are available from the corresponding author on reasonable request.” I am not at all clear what is “reasonable”, or for that matter unreasonable, request. I strongly suggest that all the data from which the findings are derived is directly available, without any need for (un)reasonable request. I believe that this is not unreasonable, but common.

We replaced this statement by “All data generated or analyzed during this study are included in this published article (and its supplementary information files)”, since we are going to upload all data of the plots in the figures (text figures as well as supplement figures) as a supplement to this manuscript.

b. “Swimming bacteria on beads”. Rephrase. It would mean that bacteria swim ON beads. One way would be to declare a composed noun, e.g., “bacterium/a-on-bead/s”, or bacterium@bead, etc.

We have rephrased it.

c. “surface consistently swim parallel to it because they become “hydrodynamically trapped” [3,4].” I believe that this is “scientific hand waving”. There is serious literature, in particular from Henry Shum (who spent all his PhD @ Oxford looking into this). He found that for mono-flagellate bacteria, which is the case here, the ratios of the cell width/cell length, and cell length/flagellum length, dictate if bacteria moved either as wall accumulator, or wall escaper, or parallel (but away) from the wall. This work was later extended (PNAS) to “all” bacterial architectures, to find out that largely this classification holds true even for more complicated architectures. It would be useful to have a discussion about it, or at the very least let the reader go deeper into Shum’s (and others) work.

We have read the paper by H. Shum et al. with interest, actually confirming the theory of hydrodynamical trapping. It is true that there are many complex behaviors, not all bacteria are swimming parallel when close to a surface and exhibit unique behaviors. However, monoflagellated bacteria, (such as *V. cholera*, which is close to *C. crescentus* in physical shape) behave as we observed, confirming that such bacteria tend to be hydrodynamically trapped when near to surfaces. We have adjusted this part and add referred to the article of H. Shum et al. for further reading on details of hydrodynamic trapping (l. 114-118).

d. Minor grammar double-check: “Once captured a bacteria on-a-bead with optical tweezers, we...”. I guess it’s “Once WE captured...”, or “Once a bacterium (singular, if “a”) is captured...”. Also, later, “Although this used approach...” does not sound like English. I think that a professional scientific editing is needed.

We have changed it.

2. Referring to the Rebuttal letter. It happens that I am not convinced that the z-staking would be that difficult. Be that as it may be, if something is difficult, it does not mean that it is not necessary (often the opposite is true), but apparently this is the authors' line of defence. If experiments are difficult, not achievable, some effort must be made to qualify the observed facts, and possibly extract inference from whatever data were available. I find that a more thorough, theoretical at least, discussion is needed here.

We discuss in the article (l. 246-252) and in our reply, that measuring movements in the z-axis would not change our interpretation, given in our setup z-movements are a very minor portion of all displacements. The reviewer does not dispute that. Moreover, does not dispute our conclusions are invalid or misinterpreted. In our view this shows that although it would show technical skill doing the measurement of z-movements, it add little to further interpretation of the biological questions we explored.

3. Recommendation. I think that at least the easy suggestions can be done without problems. The more difficult one, i.e., discussion about trajectories in 3D, could be "just" better substantiated. With these done, I guess that paper can be published.

Thank you.

Reviewer #2 (Remarks to the Author):

The authors replied to all my comments and im gladly recommending this very nice manuscript for publication!

Thank you.

In addition, owing to replotting figures S5 and S7, we made small adjustments in the main text.